# Dimethylglycine Can Enhance the Cryopreservation of Red Blood Cells by Reducing Ice Formation and Oxidative Damage

**DOI:** 10.3390/ijms24076696

**Published:** 2023-04-03

**Authors:** Yuying Hu, Xiangjian Liu, Marlene Davis Ekpo, Jiangming Chen, Xiaoxiao Chen, Wenqian Zhang, Rui Zhao, Jingxian Xie, Yongju He, Songwen Tan

**Affiliations:** 1Xiangya School of Pharmaceutical Sciences, Central South University, Changsha 410013, China; 2School of Materials Science and Engineering, Central South University, Changsha 410013, China

**Keywords:** cryopreservation, dimethylglycine, red blood cells, cryoprotectant

## Abstract

The cryopreservation of red blood cells (RBCs) holds great potential for ensuring timely blood transfusions and maintaining an adequate RBC inventory. The conventional cryoprotectants (CPAs) have a lot of limitations, and there is an obvious need for novel, efficient, and biocompatible CPAs. Here, it is shown for the first time that the addition of dimethylglycine (DMG) improved the thawed RBC recovery from 11.55 ± 1.40% to 72.15 ± 1.22%. We found that DMG could reduce the mechanical damage by inhibiting ice formation and recrystallization during cryopreservation. DMG can also scavenge reactive oxygen species (ROS) and maintain endogenous antioxidant enzyme activities to decrease oxidative damage during cryopreservation. Furthermore, the properties of thawed RBCs were found to be similar to the fresh RBCs in the control. Finally, the technique for order performance by similarity to ideal solution (TOPSIS) was used to compare the performance of glycerol (Gly), hydroxyethyl starch (HES), and DMG in cryopreservation, and DMG exhibited the best efficiency. This work confirms the use of DMG as a novel CPA for cryopreservation of RBCs and may promote clinical transfusion therapy.

## 1. Introduction

Red blood cell (RBC) transfusion is regarded as a life-saving procedure, with the primary goal of maintaining tissue and organ oxygenation in patients suffering from severe bleeding or acute anemia [1]. The American Red Cross estimates that 29,000 units of RBCs are required every day in the USA [2]. Unfortunately, RBCs for transfusion have a limited shelf life (less than 42 days) after donation due to detrimental storage effects on morphological and biochemical properties [3,4], which leads to an undesired waste of 1.7 million units of blood in the USA [5]. Therefore, prolonging RBCs’ storage duration is essential for clinical transfusion.

Cryopreservation is the use of extremely low temperatures (−80 °C or −196 °C) to preserve living cells and tissues [6,7]. It can slow or stop the metabolism of the RBCs, thus greatly extending the storage time [8]. However, there are several problems with this approach. During cryopreservation, the freezing of water harms cells in two different ways: osmotic damage and mechanical damage. First, extracellular ice formation contributes to the efflux of water from cells, and thus the cells are damaged by excessive shrinkage, which is called osmotic damage. Second, lethal intracellular ice formation (IIF) induced by the trapped water during freezing and subsequent ice recrystallization during thawing cause mechanical damage to the cells [9,10]. In recent years, oxidative damage triggered by excessive reactive oxygen species (ROS) levels has received wide attention and developed into a significant research topic [11]. Multiple effects of ROS-induced damages to cellular biomolecules can be attributed to lipid peroxidation, DNA damage, and protein oxidation [12].

Therefore, cryoprotectants (CPAs) should be added to cells to minimize damage during cryopreservation [13]. Since the first discovery of glycerol (Gly) in 1949, CPAs have received great attention [14]. Currently, Gly is the only clinically licensed CPA for cryopreservation of RBCs [15]. However, its wide application is restricted due to the toxicity of Gly and the complex deglycerolization process. Cryopreservation practices for RBCs use 20–40 wt% Gly [16]. Such high concentrations of Gly cause RBCs to undergo negative morphological changes after cryopreservation, which will further affect RBCs’ function by reducing their deformability [17]. Multiple washing steps are required prior to transfusion, which takes 30–60 min per unit (475 mL) of blood [18]. Consequently, this process does not provide immediate access to blood, which is a problem in urgent situations requiring blood transfusions [19]. Hydroxyethyl starch (HES) is an alternative to Gly for the cryopreservation of RBCs because it is safer and less toxic than Gly [20,21]. However, HES is not an effective CPA. The thawed RBCs’ recovery is less than 40%, even at high concentrations of HES [22]. Many other CPAs have been explored over the years. They include trehalose [23], betaine [24], block copolymer worms [25], and polymeric mimics of antifreeze proteins [26], etc. However, very few have shown the same efficiency as Gly. Therefore, the study for novel, efficient, and non-toxic CPAs is still ongoing.

Dimethylglycine (DMG) is a methylated derivative of the amino acid glycine with the chemical formula (CH_3_)_2_NCH_2_COOH (Figure 1) [27,28]. The molecule, first reported in 1943, is a naturally occurring intermediate metabolite in plant and animal cells [29,30]. It can be used as an antioxidant in the food industry and as a pharmaceutical intermediate to synthesize various biochemical drugs in the pharmaceutical industry [31]. Nowadays, DMG has been widely used as a dietary supplement [32], an amino acid-based surfactant [33], and a source of glycine for glutathione synthesis to improve the antioxidant capacity in the body [34].

In the current study, we have explored the potential of DMG as a novel CPA in the cryopreservation of RBCs. DMG showed satisfactory thawed RBC recovery in cryopreservation. No significant difference was found between the DMG group and the Gly group. Then, we further found that DMG could mitigate mechanical and oxidative damages during cryopreservation. The membrane properties, protein activities, and metabolite levels of thawed RBCs were also well maintained. The technique for order performance by similarity to ideal solution (TOPSIS) analyzed the advantages of DMG versus Gly and HES. These studies presented an attractive alternative to conventional CPAs and showed potential to benefit patients in clinical applications of transfusion.

## 2. Results

### 2.1. The Good Biocompatibility of DMG

Currently, the widespread use of cryopreservation is hindered by the cytotoxicity of CPAs [35]. In this work, the biocompatibility was determined by measuring the RBC recovery after incubating in various solutions at 4 °C for 48 h [7]. As shown in Figure 2, the RBC recovery in 1–4% concentrations of DMG solutions was more than 90%, indicating DMG had good biocompatibility. However, high concentrations of DMG led to relatively high hemolysis. Furthermore, we discovered that the RBC recovery in the 20% Gly group was as low as 71.89 ± 4.02%. The severe hemolysis may be caused by high osmotic pressure. Used as a clinical plasma substitute for many years, HES has excellent biocompatibility for blood [36]. The RBC recovery in 13% HES was 98.54 ± 0.72%, which is consistent with HES’s good safety.

### 2.2. Satisfactory Recovery of Thawed RBCs Cryopreserved with DMG

After the biocompatibility test, we further explored the efficiency of three CPAs to cryopreserve RBCs. The thawed RBC recovery was positively associated with the concentration of DMG within a specific range, as shown in Figure 3. The optimal concentration of DMG was 4%, and the thawed RBC recovery was up to 72.15 ± 1.22%. There was no significant difference between 4% DMG solution and 20% Gly in the thawed RBC recovery. In comparison to 20% Gly and 4% DMG solutions, 13% HES showed a low thawed RBC recovery of 42.62 ± 3.47%, indicating its poor efficiency for cryopreservation alone.

### 2.3. Ability to Protect RBCs from Mechanical Damage

Ice recrystallization inhibition (IRI) activity of DMG is evaluated by splat assay [37]. With this assay, IRI activity can be quickly assessed by monitoring the size of ice under a polarizing microscope. In the PBS group (blank control), the size of the ice was the largest (Figure 4A), which would lead to severe mechanical damage to RBCs during cryopreservation. Conversely, at concentrations of 1% to 6%, DMG had a strong IRI activity that effectively inhibited the growth of ice crystals (Figure 4B–G). Data from splat assay are commonly presented as the mean largest grain size (MLGS) (Figure 4H). In the PBS group, the MLGS reached 111.90 ± 4.82 µm. However, the MLGS was significantly reduced to 93.34 ± 4.62 µm by adding 1% DMG (*p* < 0.001). With increasing DMG concentrations, MLGS became much smaller. The results demonstrated that DMG could inhibit ice recrystallization and protect RBCs from mechanical damage. Additional cryomicroscopic images of PBS and 4% DMG that highlight ice crystal growth are displayed in Appendix A.

Differential scanning calorimetry (DSC) was used to quantitatively calculate the amount of bound water in DMG during the freezing–thawing process. In Figure 4I, the pure water sample presented a larger endothermic peak area than the DMG sample, and the proportion of bound water in the 4% DMG group was 16.56%, suggesting that DMG could produce more bound water to reduce the ice content and damage.

### 2.4. Ability to Protect RBCs from Oxidative Damage

The 1,1-Diphenyl-2-picrylhydrazyl radical (DPPH) free radical scavenging method is a simple, rapid, and accurate way to measure antioxidant activity [38]. The DPPH free radical scavenging effect of the samples can be calculated as the Trolox equivalent’s antioxidant capacity [39]. The standard curve shows a good linear relationship between DPPH clearance and Trolox concentrations with R^2^ = 0.9996 (Appendix A). DMG had the highest antioxidant capacity, reaching 20.02 ± 0.14 µgTrolox/mL. The free radical scavenging activity of different samples was in the following order: DMG > HES > Gly. PBS was not detected (Figure 5A).

RBCs express enzymatic antioxidant defense systems to prevent and alleviate intracellular oxidative stress, including superoxide dismutase (SOD) and catalase (CAT) [40]. For SOD activity, the PBS group (14.29 ± 2.37 × 10^4^ U/gHb), DMG group (16.00 ± 1.95 × 10^4^ U/gHb), Gly group (15.72 ± 1.95 × 10^4^ U/gHb), and HES group (20.41 ± 1.71 × 10^4^ U/gHb) were all higher than the control group (10.15 ± 1.59 × 10^4^ U/gHb) (Figure 5B). For CAT activity, the DMG group (30.86 ± 1.10 U/gHb), Gly (27.15 ± 1.66 U/gHb) and control group (29.11 ± 2.18 U/gHb) were similar (Figure 5C). Total CAT activity was found to be lowest in the PBS group (19.23 ± 1.67 U/gHb) and highest in the HES group (43.69 ± 3.95 U/gHb).

The oxidative damage on RBCs lipids and the effect of DMG were evaluated by measuring the level of malondialdehyde (MDA) [19]. Compared to the control group (3.16 ± 2.06 nmol/gHb), the results demonstrated that the DMG group (2.93 ± 2.73 nmol/gHb) had no effect on the lipid peroxidation level in RBCs after cryopreservation. In contrast, the PBS group (33.04 ± 4.27 nmol/gHb), Gly group (19.87 ± 0.73 nmol/gHb), and HES group (26.32 ± 3.86 nmol/gHb) induced significant lipid peroxidation (Figure 5D).

### 2.5. The Properties of Thawed RBCs

The properties of thawed RBCs were detected by the morphology, osmotic fragility, erythrocyte sedimentation rate (ESR), ATPase activities, and content of hemoglobin (Hb).

The normal morphology of RBCs is essential for their survival and oxygen-carrying capacity [41]. First, flow cytometry was used to evaluate the thawed RBCs’ morphology. In swollen RBCs, the size of forward scatter would increase, whereas in shrunken cells, the size of side scatter would increase. These analyses would show if cells are damaged in a manner that changes the shape of their surface membrane [18]. Compared to fresh RBCs (Figure 6A), the RBCs (Figure 6B) that were cryopreserved in 4% DMG showed similar profiles, suggesting their morphology is normal. For comparison, RBCs placed in a hypotonic solution of 0.7% NaCl (Figure 6C) became swollen due to water intake, and in a hypertonic solution of 2.9% NaCl (Figure 6D), RBCs shrank due to water loss, demonstrating how forward and side scatter profiles would change in response to osmotic stress. To further investigate the morphology of the RBCs, SEM analysis was performed. The fresh RBCs are normally biconcave discocytes. Many RBCs that were frozen with PBS presented acanthocytic shapes (Figure 6E), suggesting serious osmotic damage to the RBCs. Conversely, RBCs in the 4% DMG group remained regular shape (Figure 6F). Therefore, the 4% DMG solutions helped to maintain the morphology of RBCs during cryopreservation.

The ability of the cell membrane to protect structural integrity is assessed by the osmotic fragility of RBCs [42]. It should be noted that when thawed RBCs in the 20% Gly group were washed directly, the huge osmotic shock would lead to severe hemolysis. It was the reason we could not test their osmotic fragility. There was 50% hemolysis at 0.45–0.50% NaCl for fresh RBCs. The 50% hemolysis occurred at 0.60% NaCl for RBCs frozen in the 4% DMG solution and at 0.70–0.80% NaCl frozen in 13% HES, respectively (Figure 6G). Thus, although RBCs cryopreserved in 4% DMG did not have the same membrane stability as normal RBCs, their membrane stability was still superior to that of 13% HES.

ESR, one of the vital considerations of blood rheology, refers to the sedimentation velocity of RBCs in the blood. It fluctuates within a narrow range in normal RBCs and increases in many pathological states [41,43]. As shown in Figure 6H, there was no significant difference in ESR between the DMG group and the control group at 1 h, 4 h, 7 h, and 10 h. Na^+^/K^+^-ATPase and Ca^2+^/Mg^2+^-ATPase are two ATP-hydrolyzing enzymes that help maintain intracellular ion gradients. The stability of intracellular ion concentrations is important for stabilizing signal transduction and modulating cell metabolism [44]. The results showed that freezing the RBCs with 4% DMG did not alter the activities of Na^+^/K^+^-ATPase and Ca^2+^/Mg^2+^-ATPase (Figure 6I). Hb content represented oxygen carrying capacity [41], and there was no significant difference between the DMG group (19.62 ± 5.21 nmol/gHb) and the control group (21.88 ± 4.95 nmol/gHb) (Figure 6J).

### 2.6. Comparing CPAs by TOPSIS Model

TOPSIS is an effective method to deal with multiple attribute or multiple criteria decision-making problems in the real world. It is helpful for decision-makers to structure the problems that need to be resolved and to conduct analyses, comparisons, and rankings of the alternatives [45]. Hence, we used TOPSIS to quantitatively compare and assess the efficiency of Gly, HES, and DMG. The initial decision matrix was formed using three relevant criteria for CPAs, including the recovery, concentration, and biocompatibility of the thawed RBCs, as listed in Table 1. The recovery and biocompatibility of the thawed RBCs were regarded as benefit attributes, whereas concentration was the cost attribute. The results of the TOPSIS model, including the relative closeness and rank of CPAs, are displayed in Table 2. DMG had the highest closeness coefficient of 0.871, Gly and HES were 0.319 and 0.212, respectively.

## 3. Discussion

The cryopreservation of RBCs is a life-saving method that facilitates rapid access to blood samples for emergencies. However, the freeze–thaw conditions for this process are less than ideal, resulting in a low supply of RBCs. In cryopreservation, mechanical damage caused by ice formation and growth severely destroys the plasma membrane, leading to RBC death [46]. The interaction of water molecules is precisely correlated with ice formation and growth. In this study, DSC results indicated DMG could increase the ratio of bound water and inhibit the formation of ice crystals during freezing through its great water binding capacity. DMG could also greatly inhibit ice recrystallization during thawing. These effects reduced the mechanical damage to RBCs caused by large amounts of ice.

Apart from mechanical damage, oxidative damage caused by ROS is the other cause of cellular damage during cryopreservation [12]. We have investigated DMG as a protectant to prevent oxidative damage to RBCs during cryopreservation. DMG showed strong antioxidant DPPH radical scavenging activity. Moreover, the activities of endogenous antioxidant enzymes could be maintained by DMG. SOD and CAT are important enzymes that protect cells from damage caused by free radicals and ROS [47]. SOD activity in RBCs increased significantly during cryopreservation, which might be due to the fact that the stimulation of SOD activity could reduce oxidative damage and play a protective role in protecting RBCs. CAT was damaged during cryopreservation, but its activity was maintained by the addition of DMG. MDA is a lipid peroxidation product, and its accumulation has been used as an important indicator of oxidative damage brought on by ROS. The results demonstrated that DMG had no impact on the lipid peroxidation level in RBCs after cryopreservation. In contrast, both Gly and HES showed similar oxidative damage, which was seen as a higher level of lipid peroxidation after cryopreservation.

Based on the discussion presented above, the proposed mechanism of the cryoprotective effects on RBCs of DMG during cryopreservation is shown in Figure 7.

Besides recovery, the properties of thawed RBCs play a key role in transfusion therapy. We evaluated the morphology of RBCs by flow cytometry and SEM analysis, osmotic fragility, ESR, ATPase activities, and content of Hb. First, the normal morphology of RBCs is critical to their survival and function. Second, the osmotic fragility reflects the membrane stability of RBCs. Third, ESR is one of the important parameters of blood rheology. Fourth, Na^+^/K^+^-ATPase and Ca^2+^/Mg^2+^-ATPase are two ATP hydrolases that contribute to the maintenance of intracellular ionic gradients. Fifth, Hb is the main oxygen-carrying protein in RBCs. The findings confirmed that the properties of RBCs cryopreserved in DMG were well maintained, suggesting that these RBCs might be used safely.

We have used the TOPSIS model for analysis in order to quantitatively compare the performance of DMG, Gly, and HES further. The recovery, concentration, and biocompatibility of the thawed RBCs were taken into consideration to contribute to the comparison of three CPAS in the cryopreservation of RBCs. The results displayed that DMG had the best efficiency under the three criteria and showed great promise in cryopreservation, but it remains to be further investigated whether DMG can replace Gly or HES in clinical transfusion therapy.

In conclusion, we showed that DMG could work as a novel CPA and achieve a high recovery of RBCs during cryopreservation. DMG possessed a strong ability to inhibit ice recrystallization, increase the ratio of bound water, and reduce oxidative damage. After cryopreservation, the behaviors of the thawed RBCs, including their morphology, osmotic fragility, ESR, ATPase activities, and content of Hb, were found to be similar to those of the normal cells, showing that their properties were not affected. In addition, TOPSIS model analysis demonstrated that DMG had good performance in the cryopreservation of RBCs. This work offers an attractive alternative to conventional CPAs and holds great promise for the current clinical practice of RBCs.

## 4. Materials and Methods

### 4.1. Cryoprotective Solutions

Cryoprotective solutions were prepared with 20% glycerol (Sinopharm, Shanghai, China), 13% HES (Macklin, Shanghai, China), and 1%, 2%, 3%, 4%, 5%, and 6% DMG (Aladdin, Shanghai, China). All solutions were formulated with PBS (Wuhan Servicebio Technology Co., Ltd., Wuhan, China).

### 4.2. RBCs Preparation

The sheep RBCs were bought from Hongquan (Guangzhou, China). RBCs were washed with PBS in a 15 mL centrifuge tube by centrifugation (1980× *g*, 3 min). Then, the supernatant was removed. To obtain washed RBCs, the above operation was repeated twice.

### 4.3. Biocompatibility Test

To test the biocompatibility, equal amounts of washed RBCs (20 μL) were incubated in 1 mL cryoprotective solutions at 4 °C for 48 h and resuspended every 24 h. After 48 h, RBCs were collected to assess survival rates. Then, RBCs were centrifuged at 1980× *g* for 3 min. The supernatant was measured the absorbance using a microplate reader (Tecan Infinite M200 PRO, Austria) at 450 nm wavelength. A positive control was established by measuring the absorbance of the supernatant with 20 μL fresh RBCs dissolved in 1 mL deionized water, whereas a negative control was determined by adding 20 μL fresh RBCs to 1 mL PBS. The RBC recovery can be calculated using the following equation [15]:(1)$$\mathrm{Hemolysis}\;(\%)=\frac{A-A_{0}}{A_{1}-A_{0}}\times 100\%$$
(2)Recovery (%)=100%−Hemolysis (%)
where A is the absorbance of the measured sample and A_0_ and A_1_ are the negative control and positive control, respectively.

### 4.4. Cryopreservation of RBCs

Washed RBCs (20 μL) were suspended in 1 mL PBS or cryoprotective solutions as the control group and the experimental group for 20–30 min. The samples were frozen by directly immersing them in liquid nitrogen in cryogenic vials (Wuhan Servicebio Technology Co., Ltd.) for at least 20 min. Then, cells were instantly thawed in a 37 °C water bath for recovery. The thawed RBC recovery can be calculated as described above in 4.3.

### 4.5. Splat Assay

In 1986, the splat assay was established by Knight and Dumani [48]. PBS and DMG samples of different concentrations (1%, 2%, 3%, 4%, 5%, and 6%) were prepared by the above method. A 6μL droplet is dropped ~1.4 m onto a pre-cooled coverslip. The coverslip is immediately transferred to a N_2_ cooled cryostage (Huitong, LTM-190H, Shanghai, China) and annealed at −8 °C for 30 min. Afterwards, the wafers are imaged using a polarizing microscope (Huitong, XPF-550, Shanghai, China). The ten largest crystals’ sizes were measured.

### 4.6. DSC Test

The water binding capacity of DMG was investigated using DSC. The samples (10 mg) were sealed into crucibles and transferred to the calorimeter sample chamber (TA Q2000, DSC2500, USA), then cooled to −40 °C from 30 °C at a rate of 10 °C/min and heated to 10 °C at a rate of 2 °C/min. The heat flow (w/g) was monitored. The total water content (w_tc_), freezing water content (w_f_), and bound water content (w_b_) could be calculated according to the following equations [24]:(3)$$w_{\mathrm{tc}}=m_{w}/m$$
(4)$$w_{f}={\Delta H/\Delta H}_{w}$$
(5)$$w_{b}=w_{\mathrm{tc}}-w_{f}$$
where m_w_ and m are the water mass and the total mass of each sample, and ∆H and ∆H_w_ stand for the melting enthalpies of 4% DMG and pure water, respectively, which are measured by DSC during the heating process.

### 4.7. Antioxidant Assays

The antioxidant activity of DMG was estimated using the DPPH free radical scavenging method (Nanjing Jiancheng, A153-1-1, Nanjing, China). Briefly, 400 µL sample extract, or standard, and 600 µL of DPPH reagent were added and mixed vigorously. The reaction mixture was kept at room temperature for 30 min in the dark, and the discoloration of DPPH was obtained against a blank at 517 nm using the UV–Vis spectrophotometer (UV-2600, Shimadzu Europe).

Total SOD was tested by the xanthine oxidase method; the SOD activity was determined using assay kits (Nanjing Jiancheng, A001-1-1, China).

CAT activity was assessed using a commercial kit (Nanjing Jiancheng, A007-2-1, China) and quantified by analyzing the absorbance change rate of hydrogen peroxide at 240 nm [49].

The level of lipid peroxidation was quantified using 50 μL thawed RBCs lysates by the formation of the amount of malondialdehyde-thiobarbituric acid adduct in an acidic condition at 95 °C for 40 min. The absorbance of the samples was measured at 532 nm using the UV–Vis spectrophotometer (Nanjing Jiancheng, A003-1-1, China).

All the measurement methods are provided in the manufacturers’ instructions in detail.

### 4.8. Flow Cytometry

The RBC morphology was assessed by flow cytometry. Samples were made by diluting fresh RBCs in 0.9% NaCl, 2.9% NaCl, and 0.7% NaCl solutions. The experimental group consisted of post-freeze–thaw RBCs in 4% DMG solution. All samples were tested using the BD LSRFortessa™ Cell Analyzer.

### 4.9. SEM Analysis

The thawed RBCs were fixed for 12 h with 2.5% glutaraldehyde and then washed three times with PBS. Then, the RBCs were fixed once more in 1% osmic acid for 1–2 h and washed with PBS three times again. The fixed samples were dehydrated for 15 min in each gradient of 30%, 50%, 70%, 80%, 90%, and 95% alcohol before being treated twice with 100% alcohol for 20 min each. Subsequently, the RBCs were exposed to a mixture of alcohol and isoamyl acetate (*v*/*v* = 1/1) for 30 min, followed by exposure to pure isoamyl acetate for 1 h or left overnight. Next, the RBCs were dried and metalized with gold in succession. Finally, the images of RBCs were obtained using a scanning electron microscope (Hitachi, SU8010, Tokyo, Japan).

### 4.10. Osmotic Fragility

The osmotic fragility of the RBCs was determined using a stepwise dilution of 1.0% NaCl ranging from 0.1–1.0%. The thawed cells were washed twice with 0.9% NaCl at 4000 rpm to remove cryopreserve solutions. Washed RBCs (7.5 μL) were diluted 500 μL different osmotic fragility solutions at room temperature for 30 min. Afterward, the samples were centrifuged at 1980× *g* for 3 min. The absorbance of the supernatant was tested at 450 nm and the hemolysis percent of RBCs was calculated.

### 4.11. ESR Test

The erythrocyte sedimentation rate can be measured using the Westergren method, which is how quickly RBCs sink to the bottom of a blood sample [50]. After cryopreservation, a 50 μL aliquot of RBCs was added to 2 mL PBS. After being moved to a Westergren tube, the samples’ level was adjusted to the 0-scale point. The samples’ ESR was measured at 1 h, 4 h, 7 h, and 10 h.

### 4.12. ATPase Activities Assays

In accordance with the manufacturer’s instructions (Nanjing Jiancheng, A070-6-3, China), enzymatic activities assays were carried out using fresh RBCs and thawed RBCs. The RBCs were washed with 0.9% NaCl at 4000 rpm for 3 min. After removing the supernatant, DI water was added to acquire the lysis of RBCs. For the assessment of Na^+^/K^+^-ATPase and Ca^2+^/Mg^2+^-ATPase activities, the lysis of RBCs was treated with the reaction mixtures at 37 °C for 10 min. Then two reactions were stopped. The supernatant was collected to detect concentrations of inorganic phosphorus after centrifugation at 1516× *g* for 10 min [44].

### 4.13. Content of Hb

The concentration of Hb (mgHb/mL) from fresh and thawed RBCs in 4% DMG was measured by the HICN colorimetric method (Nanjing Jiancheng, C021-1-1, China). A hemocytometer was used to calculate the number of RBCs per milliliter. As a result, the content of Hb in RBCs (mgHb/10^9^ RBCs) could be acquired.

### 4.14. TOPSIS

The performance of DMG, Gly, and HES in cryopreservation was assessed using the TOPSIS model [51]. The following summarized the TOPSIS method’s application process.

Stage 1. A decision matrix was created. The decision matrix X, consisting of m alternatives and n criteria, was formed by Equation (6).
(6)$$X={\left(\chi_{ij}\right)}_{m\times n}=\left(\begin{array}{cccc} \chi_{11} & \chi_{12} & \cdots & \chi_{1n}\\ \chi_{21} & \chi_{22} & \cdots & \chi_{2n}\\ \vdots & \vdots & \vdots & \vdots \\ \chi_{m1} & \chi_{m2} & \cdots & \chi_{mn} \end{array}\right)$$

Stage 2. A normalized decision matrix was formed. Equations (7)–(9) normalized each element x_ij_ into a corresponding element S_ij_ in the matrix S.
(7)$$S={\left(s_{ij}\right)}_{m\times n}=\left(\begin{array}{cccc} s_{11} & s_{12} & \cdots & s_{1n}\\ s_{21} & s_{22} & \cdots & s_{2n}\\ \vdots & \vdots & \vdots & \vdots \\ s_{m1} & s_{m2} & \cdots & s_{mn} \end{array}\right)$$
where
(8)$$S_{\mathrm{ij}}=\frac{\chi_{\mathrm{ij}}}{\sqrt{\sum_{i=1}^{m}{{(\chi }_{\mathrm{ij}})}^{2}}},\;\mathrm{for}\;\mathrm{benefit}\;\mathrm{attribute}\;\chi_{\mathrm{ij}}$$
(9)$$S_{\mathrm{ij}}=\frac{\chi_{j}{}^{\max}-\chi_{\mathrm{ij}}}{\sqrt{\sum_{i=1}^{m}{{(\chi }_{j}{}^{\max}-\chi_{\mathrm{ij}})}^{2}}},\;\mathrm{for}\;\mathrm{cost}\;\mathrm{attribute}\;\chi_{\mathrm{ij}},\;\mathrm{where}\;\chi_{j}{}^{\max}=\underset{i}{\overset{m}{\max}}\;x_{\mathrm{ij}}$$

Stage 3. The positive ideal solutions (PIS) S_j_^+^ and negative ideal solutions (NIS) S_j_^−^ sets were constituted.
(10)$$S_{j}{}^{+}=\overset{m}{\underset{i=1}{\max}}\;S_{\mathrm{ij}}$$
(11)$$S_{j}{}^{-}=\overset{m}{\underset{i=1}{\min}}\;S_{\mathrm{ij}}$$

Stage 4. Separation measures were calculated.

In the TOPSIS method, positive ideal separation (D^+^) and negative ideal separation (D^−^) were the two separation measures for each alternative. The distance of the alternative to the PIS (D^+^)/NIS (D^−^) was determined based on Equations (12) and (13), respectively. In these calculations, the Euclidean distance method was applied.
(12)$$D_{i}{}^{+}=\sqrt{\sum_{j=1}^{n}{\left(S_{j}{}^{+}-s_{ij}\right)}^{2}}$$
(13)$$D_{i}{}^{-}=\sqrt{\sum_{j=1}^{n}{\left(S_{j}{}^{-}-s_{ij}\right)}^{2}}$$

Stage 5. Relative closeness to the ideal solution was calculated.

Equation (14) below calculated the relative closeness to the PIS of each alternative using D_i_^+^ and D_i_^−^. The alternative that is nearest to the PIS is determined to be the most suitable decision alternative.
(14)$$C_{i}=\frac{D_{i}{}^{-}}{D_{i}{}^{-}+D_{i}{}^{+}}$$

### 4.15. Statistical Analysis

Statistical analyses were performed by GraphPad Prism 9.4.1 (GraphPad Software, San Diego, CA, USA). The results of the RBC experiments are presented as the mean ± standard deviation of three independent experiments. Data were analyzed statistically by a one-way analysis of variance (ANOVA), and Tukey’s post-hoc test was used for pairwise comparisons of different groups. A *p*-value lower than 0.05 was considered statistically significant. * Indicates *p* < 0.05, ** indicates *p* < 0.01, *** indicates *p* < 0.001.

## Figures and Tables

**Figure 1 ijms-24-06696-f001:**
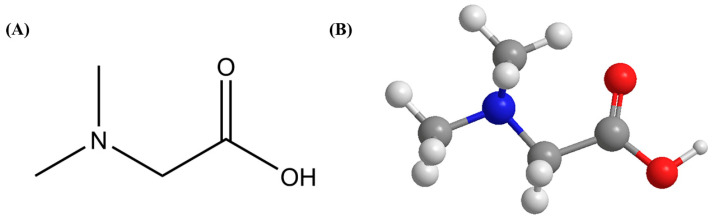
The 2D (**A**) and 3D (**B**) molecular structure of DMG.

**Figure 2 ijms-24-06696-f002:**
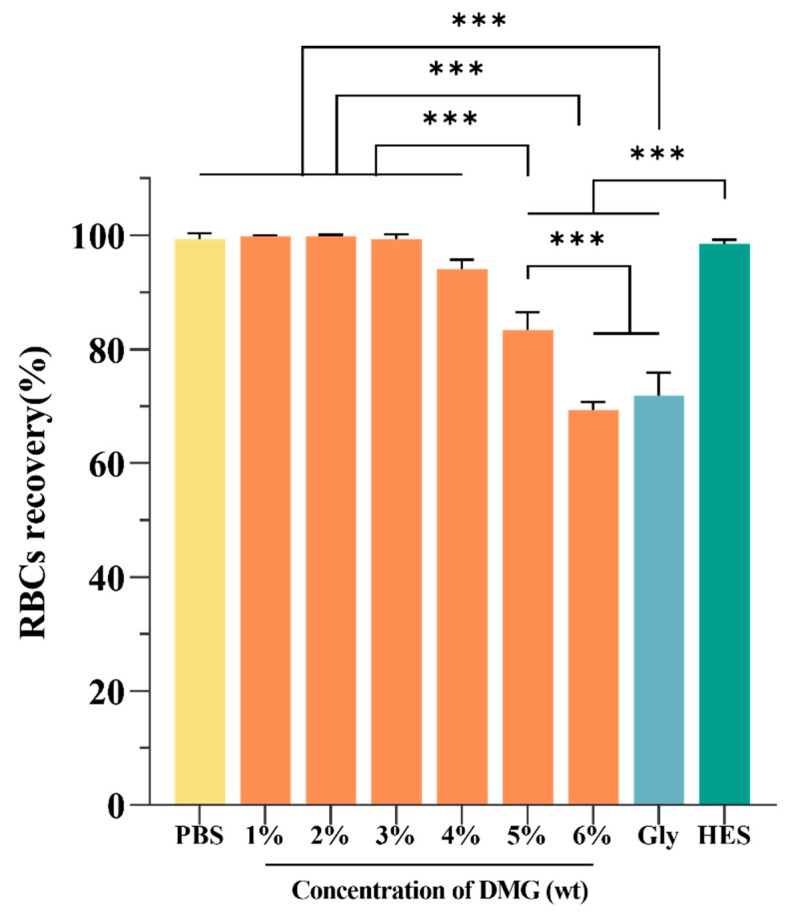
Biocompatibility test. The concentrations of Gly and HES were 20% and 13%, respectively. *** *p* < 0.001.

**Figure 3 ijms-24-06696-f003:**
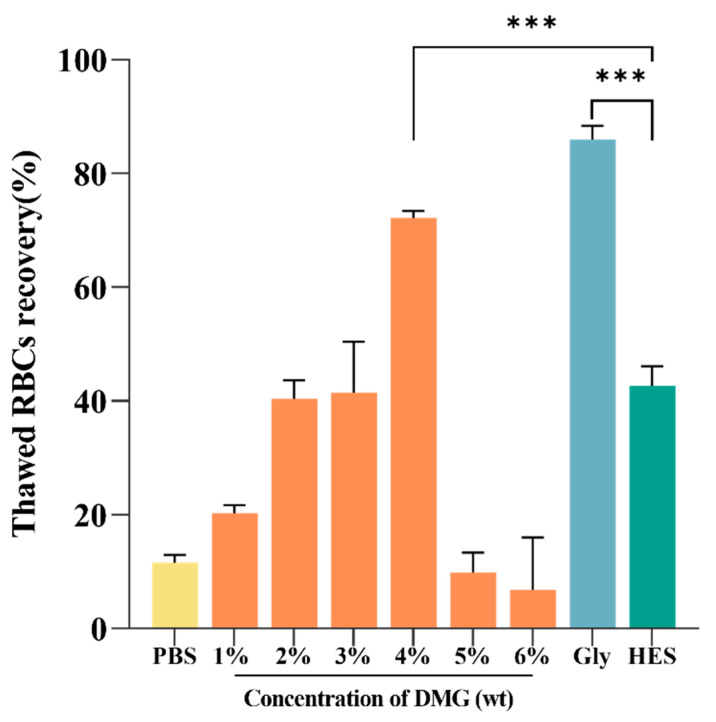
Thawed RBC recovery. The concentrations of Gly and HES were 20% and 13%, respectively. *** *p* < 0.001.

**Figure 4 ijms-24-06696-f004:**
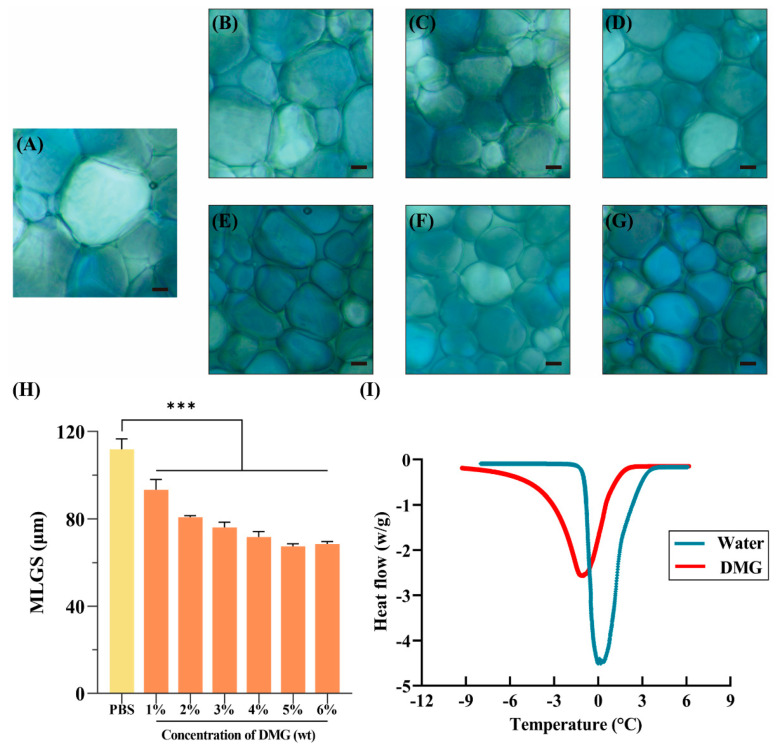
Ability to reduce mechanical damage. The representative images of ice crystals in (**A**) PBS, (**B**) 1% DMG, (**C**) 2% DMG, (**D**) 3% DMG, (**E**) 4% DMG, (**F**) 5% DMG, and (**G**) 6% DMG. (**H**) Quantitative analysis of IRI activity through mean largest grain sizes (MLGS). (**I**) The heat flow of water and 4% DMG during the melting process. Scale bar = 10 µm. *** *p* < 0.001.

**Figure 5 ijms-24-06696-f005:**
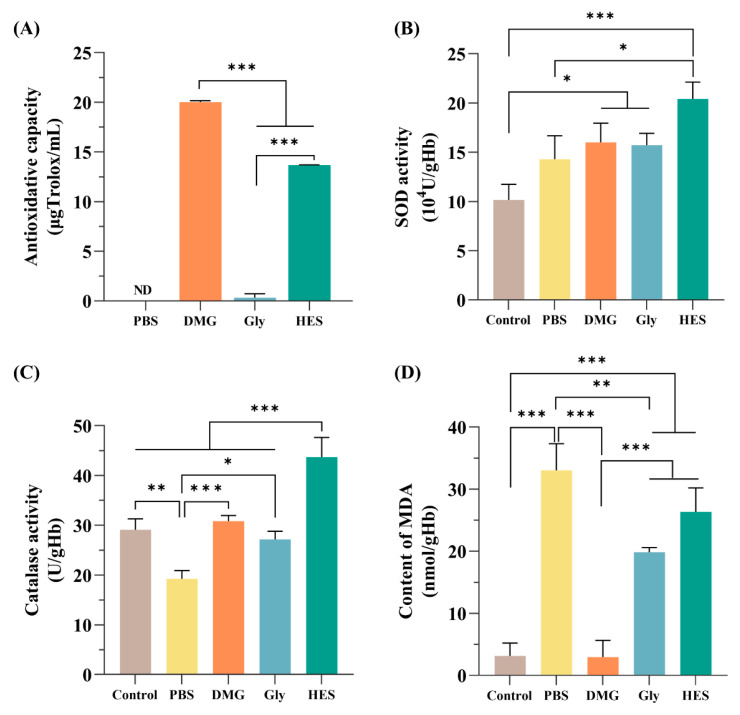
Ability to counter oxidative damage. (**A**) The antioxidant capacity was measured by the DPPH assay. The SOD (**B**) and CAT (**C**) activities of RBCs in different groups. (**D**) The content of MDA. The fresh RBCs without cryopreservation were selected as the control group. The concentrations of Gly, DMG, and HES were 20%, 4%, and 13%, respectively. Data are presented as the mean ± SD. * *p* < 0.05, ** *p* < 0.01, *** *p* < 0.001.

**Figure 6 ijms-24-06696-f006:**
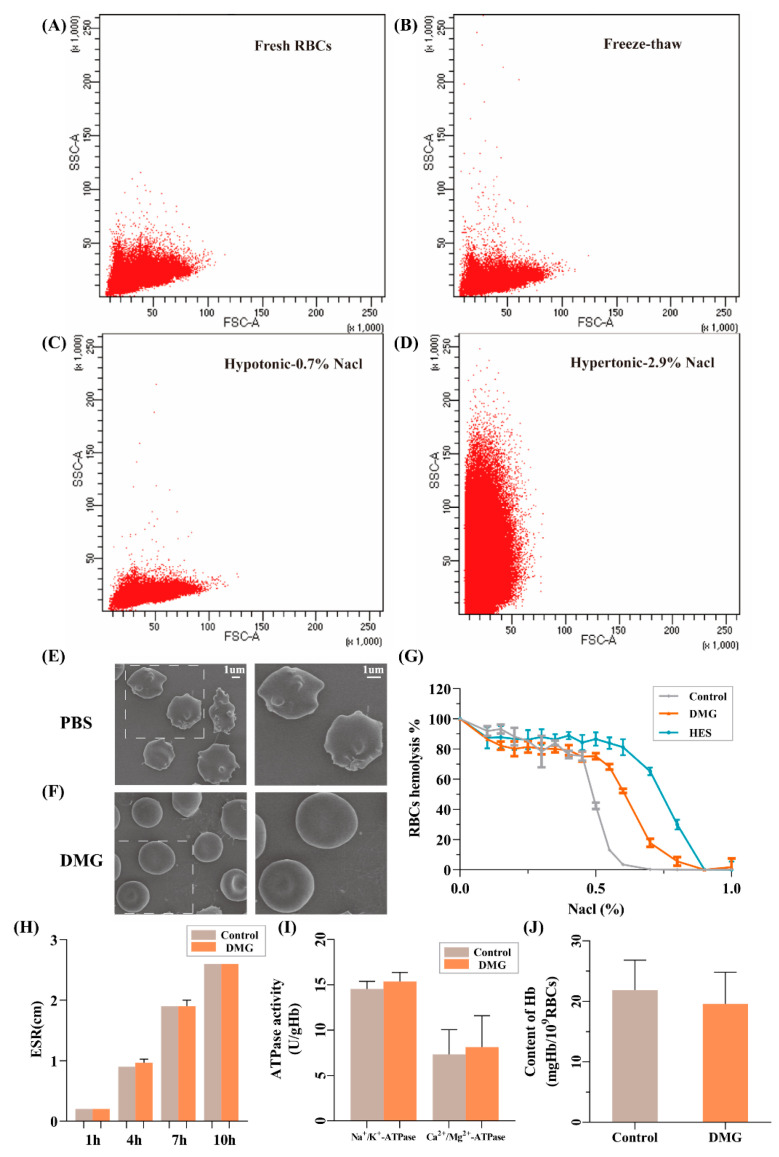
The properties of thawed RBCs after cryopreservation. Flow cytometry plots showed the forward scatter and side scatter of fresh RBCs (**A**) and post-freeze–thaw RBCs in DMG solution (**B**). For (**A**,**B**), RBCs were maintained in an isotonic environment so that they neither shrank nor swelled. For comparison, fresh RBCs placed in hypotonic 0.7% NaCl swelled due to the intake of water (**C**). In hypertonic 2.9% NaCl, fresh RBCs shrank due to loss of water (**D**). The representative SEM images of thawed RBCs incubated in PBS (**E**) and DMG (**F**). Osmotic fragility curves (**G**) of RBCs at different solutions. The ESR (**H**), ATPase activities (**I**), and content of Hb (**J**) of thawed RBCs in the control group and DMG group. The concentrations of DMG and HES were 4% and 13%, respectively. The fresh RBCs without cryopreservation were selected as the control group. Data are presented as the mean ± SD. Erythrocyte sedimentation rate (ESR), hemoglobin (Hb).

**Figure 7 ijms-24-06696-f007:**
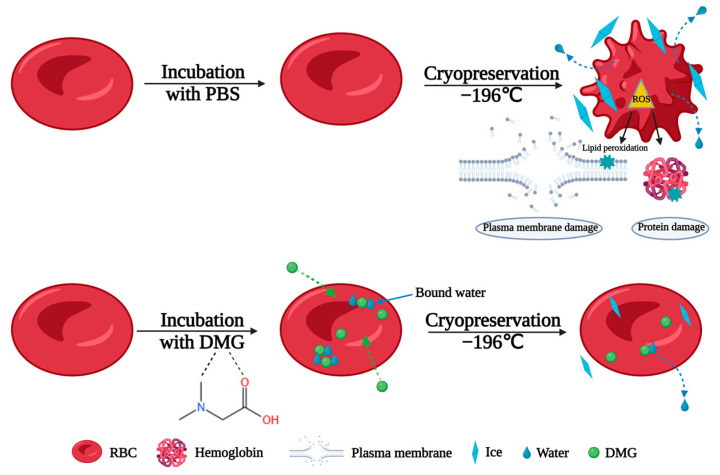
The proposed mechanism of DMG during cryopreservation.

**Table 1 ijms-24-06696-t001:** Three alternative CPAs and three criteria.

Types of CPAs	Concentration (%wt)	Thawed RBCs Recovery (%)	Biocompatibility (%)
Gly	20	85.94	71.89
HES	13	42.62	98.54
DMG	4	72.15	93.99

**Table 2 ijms-24-06696-t002:** The relative closeness and rank by TOPSIS.

Types of CPAs	Relative Closeness	Rank
Gly	0.319	2
HES	0.212	3
DMG	0.871	1

## Data Availability

All experimental data within the article are available from the corresponding author upon reasonable request.

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
