# Peer review of "Dimethylglycine Can Enhance the Cryopreservation of Red Blood Cells by Reducing Ice Formation and Oxidative Damage"

_ijms, 2023, doi:10.3390/ijms24076696_

Round 1

Reviewer 1 Report

The quality assessment & control of red blood cell(RBC) is important for safe transfusion.

(1) The authors well surveyed the literature on cryoprotectants (CPAs) for cropreservation of RBC and the background of using dimethylglycine (DMG) in the food and pharmaceutical industries.

(2) Experimental evidences showing the advantages of using DMG over other CPAs are presented including the results of flow cytometry, SEM analysis, osmotic fragility, ESR.

(3) However, experimental results of long term stability of RBC under cryopreservation conditions are essential to support their claim of using DMG in lieu of glycerol or other CPAs. Therefore, the Fig. 3, which shows the stability of RBC frozen only for a short time (~20 min) need to be replaced or supplemented by additional similar experiments with RBCs frozen for longer periods (> a few weeks, or preferably months) 

(4) The figure captions of 6E and 6F are mismatched with the corresponding text (line 220-222). 

Author Response

Dear reviewer:

       On behalf of my co-authors, we thank you very much for allowing us to revise our manuscript. We really appreciate all your positive and constructive comments and suggestions on our manuscript. The modifications in the manuscript have been marked in red. 

Reviewer 2 Report

In the article by Hu et al. titled “Dimethylglycine Can Enhance the Cryopreservation of Red

Blood Cells by Reducing Ice Formation and Oxidative Damage”, authors reported the results of the study of a new CPA, dimethylglycine, for the cryopreservation of RBCs.

In my opinion the authors carried out good structured experiments to assess the effectiveness of this excipient for RBCs preservation.

In general, the manuscript is well structured.

Regarding the results, they are not presented using the Taylor’s rules for the stating of the results. For example, 71.89 ± 4.02% becomes 72 ± 4%. Please correct them in the whole manuscript.

I have also some minor comments:

In the abstract there are too many acronyms not used (ROS, TOPSIS and Gly), while HES is not defined.

Line 36: the symbol of Celsius degrees is not correct in the whole manuscript.

Figure 1: Is the figure done by the author? If not, or partially taken from other sources, authors have to add the references.

Figure 2: Error bars have been trimmed. Please provide a complete figure with a y axis starting from 0 to a value that allows the visualization of all the error bars.

Figure 3: In my opinion it is not necessary indicating on the graph the not significant comparisons, but only the significant ones. You can specify the not significancy in the main text.

Figure 4: I think it is not necessary to specify the name of the samples on the figure since they are already specified in the caption. If you remove the titles, you can also enlarge the ice crystals images, made them clearer. As for figure 2, start the y axis of the graph H from 0.

Figure 6: Increase the font dimensions of panels A-D and the scale bars of E and F. Why there are dotted lines on images? I guess it was to indicate the enlargements, but I think that the oblique one is useless.

Table 1: Please format the table to not exceed main text dimensions

Lines 372-375: why did you choose those concentrations? Could you provide some references and/or motivation of your choice?

Lines 378-381: I don’t understand the procedure. Did you wash the RBCs two or three times? I think this passage is not clearly described.

Line 384: remove the capital letter to “Equal”

Line 385: What do you mean for resuspended? I think that if the total incubation time is 48h it is sufficient to indicate after 24h instead of every 24 hours.

Line 442: the acronym MDA-TBA is not further used

Line 487: TCA is not defined

Figure S1: The scale bar is missing on images

Figure S2: I cannot understand the y axis. The unit is %, but the maximum is 100% or 1%?

Author Response

(The authors gave the same response as above.)

Round 2

Reviewer 1 Report

Although I find a few papers where longer times, overnight (Carpenter et al., Proc. Natl. Acad. Sci. 1992) and 5 days (Mitchell et al., ChemComm. 2015) of cryopresevation, were used for a similar study, I now agree with the authors, based on their answers to my previous comment, in saying that probing the efficient inhibitory action of DMG to water-to-ice phase transition during the freezing and thawing process is enough to support their claim of DMG as a novel CPA.